

# Volume, formation and sedimentation of future glacier lakes in Switzerland

Tim Steffen[1,2], Matthias Huss[1,2,3], Rebekka Estermann[1,2], Elias Hodel[1,2], Daniel Farinotti[1,2]

[1]Laboratory of Hydraulics, Hydrology and Glaciology (VAW), ETH Zurich, Zurich, 8092, Switzerland
[2] Swiss Federal Institute for Forest, Snow and Landscape Research (WSL), Birmensdorf, 8903, Switzerland
[3] Department of Geosciences, University of Fribourg, Fribourg, 1700, Switzerland

*Correspondence to*: Daniel Farinotti (daniel.farinotti@ethz.ch)

**Abstract.** Ongoing climate change and associated glacier retreat is causing rapid environmental change, including shifts in
high-alpine landscapes. Glacier lakes, which can form in topographical depressions left behind by glacier retreat, are prominent features within such landscapes. Whilst model-based estimates for the number and area of future glacier lakes exist for various mountain regions across the world, the exact morphology and temporal evolution remain largely unassessed. Here, we leverage a recently released, measurement-based estimate for the subglacial topography of all glaciers in the Swiss Alps, to provide an estimate about the number, size, time of emergence, as well as sediment infill of future
glacier lakes. The topographical information is based on 2,450 km measured ice thickness profiles, whilst the temporal evolution of glaciers is obtained from a glacier evolution model forced with an ensemble of climate projections. We estimate that up to 683 potential lakes with an area >5,000 m$^2$ and a depth >5 m could emerge across the Swiss Alps if glaciers were to disappear completely, with the potential to hold a total water volume of up to 1.16 [1.05, 1.32] km$^3$ (numbers and 95% confidence interval). For a middle-of-the-road climate scenario, we estimate that about 10% (0.12 [0.04,
0.18] km$^3$) and 48% (0.56 [0.26, 0.67] km$^3$) of this volume could be realized by 2050 and 2100, respectively. In a first-order assessment, we also estimate that ca. 45% of the newly emerging glacier lakes (260 out of 570) will be transient features, i.e. will disappear again before the end of the century owing to refilling with sediments released by glacial erosion and proglacial sediment transport.

**1 Introduction**

As global temperatures continue to rise, worldwide mountain glaciers are rapidly shrinking (Hock et al., 2019). Also glaciers in the Alps are affected, with further loss of glacier volumes being inevitable (Marzeion et al., 2018; Zekollari et al., 2020). Depending on climate scenario, the Alps may lose up to 94% of their 2020 ice volume by the end of the 21$^{st}$ century (Zekollari et al., 2019), with important consequences for the appearance of landscapes as such (Orlove et al., 2008).
Retreating glaciers produce environments dominated by erosion and deposition, consisting of hills, sinks, and overdeepenings amongst other (e.g. Cook and Swift, 2012). Overdeepenings, i.e. confined topographical depressions caused by erosion, may entirely replenish with sediments transported by glaciers and their proglacial streams, or may fill



with water giving rise to new glacier lakes (Frey et al., 2010; Mölg et al., 2021). Predicting which of the two scenarios will materialize is difficult, since the spatio-temporal dynamics of glacial erosion is governed by a complex interplay of

processes that are difficult to quantify and, thus, to anticipate (e.g. Lane et al., 2017). In their recent review, Carrivick and Tweed (2021) highlighted that at the worldwide scale, specific sediment yields from glacierized catchments can span as much as five orders of magnitude, with sediment yields in the European Alps ranging from dozens to thousands of tons per year and square kilometre. Moreover, the sediment yields are strongly controlled by glacierization itself (Hinderer et al., 2013), adding a significant temporal dependence on the corresponding estimates.

Despite the difficulty in predicting their emergence, the interest in potential new glacier lakes has recently been on the rise. Indeed, such lakes have been identified to pose both risks and opportunities (Haeberli et al., 2016; Anacona et al., 2018). On the one hand glacier lakes represent a potential hazard for downslope populations and infrastructure if they burst (Frey et al., 2010; Emmer et al, 2014; Anacona et al., 2018), with a number of studies aiming at identifying already-existing hazardous lakes (e.g. Bolch et al., 2012; Huggel et al. 2002; Veh et al., 2018; Zhang et al., 2022) and at clarifying whether

any change in the frequency of their outbursts can be detected at large scales (e.g. Veh et al., 2019). On the other hand, new glacier lakes are of relevance for high-mountain biodiversity (Čiamporová-Zaťovičová and Čiampor, 2017; Tiberti et al., 2019), have been shown to hold significant hydropower potential (Ehrbar et al., 2018; Farinotti, et al., 2019), might be attractive to tourists (Purdie, 2013; Welling et al., 2015), or might serve as reservoirs for artificial snow production in ski areas (Haeberli et al., 2016) and other water management purposes (Farinotti et al., 2016; Brunner et al., 2019).

In light of the above relevance, accurate knowledge about the size, the distribution, and the time of formation of glacier lakes is key. At the worldwide scale, there are multiple examples of inventories of already-existing glacier lakes (e.g. Komori, 2008; Gardelle et al., 2011; Zhang et al., 2015; Emmer et al., 2016; Petrov et al., 2017; Drenkhan et al., 2018; Shugar et al., 2020; Wang et al., 2020), with most studies concurring that the formation of glacier lakes has been accelerating during the past decades (see also the review by Carrivick and Tweed, 2013). In the European Alps, inventories

of existing proglacial lakes are available for Austria (Emmer et al., 2015; Buckel et al., 2018), various parts of Italy (Galluccio, 1998; Salerno et al., 2014; Viani et al., 2016), and Switzerland (Mölg et al., 2021). According to the latter, Switzerland hosted 987 proglacial lakes in 2016, covering an area of $6.22 \pm 0.25$ km$^2$ in total. Additional 205 glacier lakes have been fully replenished with sediments since 1850 and have thus disappeared again (Mölg et al., 2021). These numbers show that the emergence and disappearance of glacier lakes is a dynamic process, and calls for anticipating the location and

timing of future glacier lakes as well.

Several studies aiming at such an anticipation of future glacier lakes exist, notably for (parts of) High Mountain Asia (Linsbauer et al., 2016; Kapitsa et al., 2017; Zheng et al., 2021), the Andes (Colonia et al., 2017; Drenkhan et al., 2018), or the European Alps (Linsbauer et al., 2012; Magnin et al., 2020; Viani et al., 2020; Gharehchahi et al., 2020). For the Swiss Alps, Linsbauer et al. (2012) estimated that between 400 and 600 new glacier lakes could form if glaciers were to vanish

entirely, with a total area in the order of 50 to 60 km$^2$ and a total volume of about 2 km$^3$. Gharehchahi et al. (2020), who



focused on the Swiss part of the Rhone basin, anticipated up to 171 potential new glacier lakes in this area, with a total volume of about 0.5 km$^3$. Such studies are based on estimates of the subglacial topography and the assumption that any present-day subglacial overdeepening will form a lake once the glacier will have retreated from the corresponding location. The latter assumption ignores any topography changes that could emerge from sediment erosion and deposition, whilst the

subglacial topography is inferred by subtracting a spatially-distributed estimate of the glacier ice thickness from contemporaneous information of the glacier surface topography. Since direct measurements of glacier ice thickness are generally sparse (Welty et al., 2020), glacier-wide ice thickness distributions are typically inferred with the help of inversion models (for a review , see Farinotti et al., 2017). The accuracy of such models has been assessed in two recent intercomparison experiments (Farinotti et al., 2017; 2021) with the results highlighting that such models are indeed skilful

in estimating the glacier-wide ice thickness, but that point-based estimates can suffer from considerable uncertainties when at a certain distance from direct measurements. For glaciers with such measurements, the Ice Thickness Models Intercomparison eXperiment Phase 2 (ITMIX2; Farinotti et al., 2021) estimated a point-based uncertainty in the order of 16% of the mean glacier thickness. Clearly, this uncertainty directly affects the estimates for the location and size of potential future glacier lakes, and indicates that extensive ice thickness surveys are necessary for estimating the

characteristics of potential future lakes with some confidence.

In this study, we rely on such extensive ice thickness surveys to present a new estimate for the potential formation of future glacier lakes in the Swiss Alps. More specifically, we rely on the Swiss-wide subglacial topography recently released by Grab et al. (2021) on the basis of almost 2,500 km of ground penetrating radar (GPR) surveys, and use it to detect the location and size of subglacial overdeepenings that could give rise to glacier lakes after their retreat. In contrast to previous

studies, we also quantify the timing of the potential lake formation. We do so by combing the ice-free subglacial topography with results from the Global Glacier Evolution Model (GloGEM; Huss and Hock, 2015) forced by state-of-the-art climate model projections. This provides insights into the water volumes that could be retained in future glacier lakes under different climate scenarios. For the first time, we also aim at roughly quantifying future sedimentation rates which affect the overdeepenings after glacier retreat, thus providing indications for the long-term persistence of the emerging

lakes. The resulting estimates for the temporal evolution of future glacier lakes provide a first glimpse on how Alpine landscapes might change throughout the 21$^{st}$ century.

## 2 Study region and data

The geographical extent of this study is given by the Swiss Glacier Inventory 2016 (SGI2016; Linsbauer et al., 2021),

which can be divided into the four large river catchments of Switzerland, i.e. Rhine, Rhone, Po, and Inn (Fig. 1). The SGI2016 is an inventory of all Swiss glaciers which has been produced on the basis of high-resolution aerial images acquired between 2013 and 2018 (centre year: 2016). The 1,400 inventoried glaciers cover an area of 961 km$^2$. Most



glaciers are small, with only 16 glaciers being larger than 10 km$^2$. Eleven of the latter are situated in the Rhone river basin (Fig. 1) as it hosts the highest peaks of the Swiss Alps. The surface topography of each glacier is taken from the digital
elevation model (DEM) swissALTI3D (Swisstopo, 2019), which refers to the year 2015 on average. For this study, SwissALTI3D has been resampled to a 10m spatial resolution (the native resolution is 2m).

The subglacial topography of all considered glaciers is taken from Grab et al. (2021). This topography is based on both extensive helicopter-borne GPR measurements (Rutishauser et al., 2016; Langhammer et al., 2019a, 2019b) and ground-based GPR data. In total, 2,450 km of GPR profiles collected on 251 different glaciers and covering 81% of the SGI2016
glacier area were available (cf. Table 1 in Grab et al, 2021). Since the density of the GPR measurements varies between glaciers and since direct ice thickness measurements are not available for all glaciers, Grab et al. (2021) used two different ice thickness models to obtain a spatially complete estimate: the *Glacier Thickness Estimates* model (GlaTE; Langhammer et al., 2019a) and the *Ice Thickness and Volume Estimation based on Observations* model (ITVEO; Huss & Farinotti, 2012). Both of these models are based on principles of ice flow dynamics and are designed to optimally make use of the
information contained within sparse measurements of ice thickness. The performance of both models has been assessed within ITMIX2 (Farinotti et al., 2021) and since the average of several ice thickness models has been shown to yield most robust results (Farinotti et al., 2017), the final dataset was obtained by averaging the ice thicknesses estimated by GlaTE and ITVEO. Henceforth we will refer to the subglacial topography obtained by subtracting this average ice thickness from the SwissALTI3D surface topography as to the "mean bedrock topography". The spatial resolution of this topography is of
10m, whilst an estimate for the local vertical accuracy (based on the distance to the next GPR measurement and the difference between the estimates of GlaTE and ITVEO) is provided as separate information to the dataset. For further details on the methodology, refer to Grab et al. (2021).

Our detection of potential lakes within the glacier extent of the SGI2016 is based on the abovementioned mean bedrock topography. Following previous studies, we define potential lakes through the detection of overdeepenings in the subglacial
topography, i.e. we only consider bedrock-dammed lakes and neglect potential ice-dammed lakes or lakes dammed by moraines not resolved by the GPR data. The overdeepenings are detected by applying the tool "Fill" from the ArcGIS "Hydrology Toolset" (Esri, 2020) to the mean bedrock topography. This operation yields a spatially distributed dataset in which every overdeepening within the currently glacierized area is filled. We assume the difference between this dataset and the original mean bedrock topography to represent the depth of potential future glacier lakes, and detect their extents by
drawing polygons around connected areas with depth >0 m. The latter operation is performed by using the ArcGIS tool "Raster to Polygon (Conversion)" (Esri, 2020). Only polygons with an area >5,000 m$^2$ and a maximal depth >5 m are retained for further analysis. These thresholds were arbitrarily set and are meant to remove small-scale features that lie within the uncertainty of the underlying bedrock.

Being constrained to the area within the SGI2016 glacier outlines, the above procedure fails to identify already-existing
glacier lakes that are in contact with glacier ice and that might further expand in future (one of the most prominent



examples is the lake presently in front of Rhonegletscher, at the source of the Rhone river). To include such cases in our analyses, we visually inspected aerial images from Swisstopo (Swisstopo, 2021). We detected 15 lakes that are presently in contact with ice but are dammed by rock or sediment. To include them, we manually extended the lake polygons obtained with the procedure described above as. Since the bathymetry of the so-added lake portions is generally unknown, we

pragmatically assumed their depth to be equal to the mean depth of the remaining lake portion (i.e. the lake portion that is presently glacierized and that has a depth estimate based on the available ice thickness information).

To distinguish between individual lakes, a lake identifier (lake-ID) is defined. The lake-ID is composed of the SGI-ID (i.e. the glacier identifier of the SGI2016) followed by the rank of the lake volume within the particular glacier. That is: the largest lake of the glacier with SGI-ID "B40-07" (that is Fieschergletscher), is named "B40-07-01", the second largest is

named "B40-07-02", and so on. This nomenclature is used throughout the article and in the data that we provide as digital supplement (see section "Data availability").

## 3.2 Glacier retreat and timing of lake formation

For estimating the timing of future lake formation, we rely on glacier retreat projections scenarios based on GloGEM (Huss

and Hock, 2015). The model describes the main processes determining glacier surface mass balance (snow accumulation, ice melt, refreezing) and computes annual surface elevation change – and thus glacier retreat or advance – based on a mass-conserving parameterization (Huss et al., 2010). The mean bedrock topography by Grab et al. (2021) is used for determining the current ice thickness distribution, and the model has been applied to all individual glaciers of the SGI2016. GloGEM has been calibrated to glacier-specific observations of ice volume change between 2000 and 2019 (Hugonnet et

al., 2021). For computational reasons, the model is discretized into 10m elevation bands but results on area and thickness changes in individual bands are extrapolated to the same 10mx10m grid that is used in the analysis of potential glacier lakes (cf. Sec. 3.1). The model is forced with gridded monthly temperature and precipitation data from the ERA-5 re-analysis (Hersbach et al., 2020) for the past, and with results from a total of 56 different Global Circulation Model (GCM) runs or the future and until the end of the 21$^{st}$ century. The 56 runs stem from 13 different GCMs used in the sixth phase of the

Coupled Climate Model Intercomparison Project (CMIP6; Eyring et al., 2016) and are based on five different Shared Socio-economic Pathways (SSPs; Meinshausen et al., 2020; note that for SSP119 only 4 GCM runs are available). The latter describe future climate forcing due to both different greenhouse gas emissions and political environment. Whereas SSP119 and SSP126 refer to low-emission scenarios assuming global $CO_2$-neutrality by the second half of the century, SSP585 is a high-emission scenario with limited efforts to mitigate climate change. In the following, results of the

intermediate scenario SSP245 is used in most analyses to illustrate the results.

For each of the 56 climate model chains considered and for each potential glacier lake, we compute the year when >0%, 25%, 50%, 75% and 100% of the lake area become ice-free, as well as the yearly lake volume for the area that has become





ice-free. In the following, results are aggregated by SSPs, providing (i) the average year of lake formation, (ii) a lake volume time series for each SSP, as well as (iii) an uncertainty range given by the spread in the results obtained from different GCMs forced with the same SSP.

### 3.3 Lake sedimentation

In order to obtain a first-order estimate for the time required to fill overdeepenings becoming ice-free with sediments, we propose a simple approach that is tightly connected to the GloGEM results. Accounting for the spatio-temporal dynamics of sedimentation of new glacier lakes is crucial as (1) many – especially smaller – overdeepenings will be sedimented a few years after their formation (e.g. Mölg et al., 2021), and (2) erosion and sediment-transport rates in glacial environments are known to be extreme (for compilations of such rates, see Hinderer et al., 2013 or Carrivick and Tweed, 2021). In this respect, glacial overdeepenings represent important sediment traps (e.g. Geilhausen et al., 2013; Bogen et al., 2015), and the connectivity of the fluvial system emerging after glacier retreated must be considered since lower-lying areas might be deprived of sediment input once new traps come into existence above (Micheletti and Lane, 2016; Lane et al., 2017). A detailed description of the sediment processes that affect deglacierizing areas is out of reach for regional to global-scale glacier models, and considerable simplifications are necessary. The approach described below attempts to quantify the most relevant drivers of sediment yields into proglacial basins, and to capture the corresponding spatio-temporal dynamics.

We parameterize the sediment volume transported into a glacial lake via (i) the sediment load per unit volume of water ($c_{\text{sed,in}}$, in kg m$^{-3}$), and (ii) the glacial runoff (in m$^3$ s$^{-1}$) originating from the catchment area above the considered lake in month $m$ ($Q_{m,\text{top}}$) according to GloGEM (Huss and Hock, 2018). Furthermore, we consider the typically much lower sediment concentrations in water flowing out of a proglacial lake ($c_{\text{sed,out}}$), i.e. after sedimentation (e.g. Bogen et al., 2015). Monthly sediment volumes deposited in a proglacial overdeepening are thus computed as

$$V_{sediment,m} = \frac{c_{sed,in} - c_{sed,out}}{\rho_{sed}} \cdot Q_{m,top} \,, \tag{1}$$

where $\rho_{\text{sed}}$ is the estimated density of deposited sediment, here set to 2200 kg m$^{-3}$ (Hinderer et al., 2013). For simplicity, we assume sediment output from the lake $c_{\text{sed,out}}$ to remain constant in time, and set it to 0.05 kg m$^{-3}$ based on literature values (e.g. Hinderer et al., 2013). As for $c_{\text{sed,in}}$, we are aware that $c_{\text{sed,out}}$ may be subject to temporal variations but we assume these to be dampened by the lakes and to be relatively small in absolute terms. In our approach, $c_{\text{sed,in}}$ will be subject to major variations depending on the sediment sinks above the considered lake, as well as due to the erodibility of the basin which is influenced by factors such as slope, recently exposed proglacial area, and the height of the headwall (Ballantyne, 2002; Benn and Evans, 2010; Costa et al., 2018; Antoniazza and Lane., 2021; Carrivick and Tweed, 2021). In order to estimate $c_{\text{sed,in}}$, we parameterize three main processes: (1) subglacial abrasion, (2) the increase in deglacierized area, and (3)



glacial and periglacial erosion. In the following we describe how these processes are quantified individually, noting that our approach is highly parameterized and simple.

(1) Subglacial abrasion scales with glacier flow speed (e.g. Herman et al., 2015), and is parameterized as a non-dimensional index $i_{abrasion}$ depending on instantaneous mean glacier thickness $h_{mean}$ and mean glacier slope $\alpha_{mean}$:

$$i_{abrasion} = \frac{h_{mean} \cdot \alpha_{mean}}{h_{crit} \cdot \alpha_{crit}}, \qquad (2)$$

where $h_{crit}$ and $\alpha_{crit}$ are critical values for mean thickness and slope that correspond to an average Swiss glacier (e.g. Linsbauer et al., 2021). For thick or steep glaciers, $c_{sed,in}$ is thus increased relative to the reference value, and vice versa for thin or flat glaciers. No contribution from this process occurs once the glacier has completely disappeared.

(2) The deglacierized area is known to be a major erosion source due to exposed unconsolidated sediment (e.g. Delaney et al., 2018b). Analogously to (1), we parameterize this effect with an index $i_{proglacial}$ dependent on the area $A_{proglacial}$ exposed by glacier retreat since the beginning of the simulations, its average age $t_{proglacial}$, and its average slope $\alpha_{proglacial}$:

$$i_{proglacial} = \frac{t_{crit}}{t_{proglacial}} \cdot \frac{\alpha_{proglacial}}{\alpha_{crit}} \cdot \frac{A_{proglacial}}{A_{basin}}, \qquad (3)$$

Here, $t_{crit}$ is the full time interval of the simulations (i.e. 100 years, from 2000 to 2100), and $A_{basin}$ is the total area of the basin. Thus, a large, steep and recently deglacierized proglacial area will result in higher $c_{sed,in}$ than a small, flat, or long-established one. Geometrical parameters for the shape and the age of the proglacial area are extracted from GloGEM results at annual time steps.

(3) Glacial and periglacial erosion is most powerful in headwalls (MacGregor et al., 2009). For each glacier, we compute the area and the average slope of the headwall, here defined as the non-glacierized area above the glacier's median elevation lying within the glacier's hydrological catchment (the latter is defined based on the DEM extending beyond the presently glacierized surfaces). The index $i_{headwall}$ depends on both the headwall area $A_{headwall}$ relative to the original glacier area and the mean slope of the headwall $\alpha_{headwall}$:

$$i_{headwall} = C_{headwall} \cdot \frac{A_{headwall}}{A_{basin}} \cdot \left(\frac{\alpha_{headwall}}{\alpha_{crit}}\right)^2, \qquad (4)$$

where $c_{headwall}$ is a dimensionless scaling parameter empirically set to 2.5. The square in the right-most term of the equation is meant to qualitatively capture the exponential effect that headwall slope has on the erosion potential, although further studies would need to confirm this relation quantitatively. Finally, the indices for the three considered factors are averaged to yield $c_{sed,in}$:

$$C_{sed,in} = C_{sed,in\ ref.} \cdot \frac{(i_{abrasion} + i_{proglacial} + i_{headwall})}{3}, \qquad (5)$$



where $c_{sed,in\ ref.} = 0.50$ kg m$^{-3}$ is a reference sediment concentration in proglacial runoff for any basin, the value being
determined based on long-term observations in proglacial streams in the Swiss Alps (e.g. Hinderer et al., 2013; Delaney et
al., 2018a). As the three indices $i_{abrasion}$, $i_{proglacial}$ and $i_{headwall}$ are annually updated based on the basin's current morphology,
also $c_{sed,in}$ will mirror long-term variations in the basin's characteristics. In particular, $c_{sed,in}$ will respond to changes in
glacier ice extent, the condition of the proglacial area, and the extent of the headwall. Albeit empirical, the approach allows

for capturing both the high variability in erodibility of different glacier catchments, as well as the temporal dynamics of
sediment yield in the transition from glacierized to ice-free basins.

In the case that multiple potential glacier lakes are exposed at the same time, we assume all of them to be directly linked
with each other, and to transfer water and sediments from the higher lakes to lower ones. Two-dimensional aspects of
connectivity are neglected for simplicity. The topmost exposed proglacial overdeepening receives a sediment yield

corresponding to Eq. (1), computed using all runoff originating from above the respective elevation. Lower-lying potential
lakes, however, are only fed by sediment yields computed based on runoff from the elevation interval up to the next higher
potential lake, which typically results in much smaller sediment input. For each potential lake, the volume that is still free
from sediments is annually updated, and as soon as it is completely sedimented, it is no longer considered as a sediment
sink. This can then cause the sediment yields to rise again in lower-lying lakes.


### 3.4 Uncertainties

Our results are affected by various uncertainties, including uncertainties in the lakes' morphology (i.e. location, total area,
and total volume) and temporal evolution (i.e. year of formation and rate of sedimentation). The following paragraphs
describe how each of these uncertainties is estimated.

### 3.4.1 Uncertainty in lake location and area

The location and areal extent of the individual, potential lakes are determined by the subglacial topography. This means
that in general, the number of lakes, the lake extents, as well as the lake locations computed on the basis of the subglacial
topography generated with GlaTE (cf. Sec. 2) might be different from the ones computed by using the subglacial
topography generated with ITVEO. Similar is true when comparing the lakes computed on the basis of the mean bedrock

topography with either of the results from GlaTE or ITVEO. Because of these differences, it is not possible to establish a
one-to-one relation between lakes generated on the basis of the individual bedrock topographies and we thus estimate the
uncertainty in lake extents by aggregating the results at the level of individual glaciers. More specifically, we compute (1)
the total area of lakes obtained for a given glacier by using the GlaTE bedrock topography and (2) the total area of lakes
obtained for the same glacier by using the ITVEO topography, and use the relative difference of these totals ($\Sigma A_{GlaTE}$ and

$\Sigma A_{ITVEO}$, respectively) as an estimate for the relative uncertainty ($\sigma_A$) of each lake with area $A$, i.e.



$$\frac{\sigma_A}{A} = \frac{|\sum A_{GlaTE} - \sum A_{ITVEO}|}{avg(\sum A_{GlaTE}, \ \sum A_{ITVEO})}. \tag{6}$$

This uncertainty is assumed to apply to every lake, notably also to the lake estimated with the mean bedrock topography. We also assume the individual lake extents to be independent, meaning that the uncertainty for the total area of a given set of lakes is computed by adding the uncertainty of individual lakes in quadrature.

### 3.4.2 Uncertainty in lake volume:

Also the uncertainty in lake volume is controlled by the uncertainty in the subglacial topography. The dataset by Grab et al. (2021) provides information about the local uncertainty of the bedrock elevations ($z$), and does so by providing an upper and a lower bound ($z^+$ and $z^-$, respectively) for the corresponding values. These bounds are based on an analysis of the used interpolation methods, the uncertainty in the GPR profiles, and the uncertainties of the surface DEM. In general, the uncertainty in the subglacial topography increases with the distance to the closest GPR measurement (for details, refer to

Grab et al., 2021).

Since the total volume of each lake ($V$) is given by integration of all lake depths within the lake's area ($A$), and since the lake depth information is provided on a grid with grid-cell area $A_{cell}$, we use the uncertainty-estimates provided by Grab et al. (2021) to compute an upper and a lower bound for the lake's total volume ($V^+$ and $V^-$, respectively):

$$\begin{cases} V^+ = V + A_{cell} \cdot \sum(z^+ - z) \\ V^- = V + A_{cell} \cdot \sum(z^- - z) \end{cases}. \tag{7}$$

In the equation, $V$ is the lake volume computed by using the mean bedrock topography, and the sums are performed over all

grid-cells that lie within the area of the given lake. Note that, by definition, $z^+ \geq z \geq z^-$ (and thus $V^+ \geq V^-$), and that in general $|z^+ - z| \neq |z^- - z|$. The latter leads to an asymmetric confidence interval around $V$.

Since we don't expect the results by Grab et al. (2021) to be systematically biased towards either $z^+$ or $z^-$, we treat the uncertainties in different lake volumes as independent and thus sum the estimated uncertainties in quadrature.

### 3.4.3 Uncertainty in the year of lake formation

We estimate the uncertainty in the year of lake formation as the standard deviation of the 13 different years of lake formation resulting from the 13 GCMs considered for each SSP. The year of lake formation will in fact also be affected by other uncertainties such as the subglacial topography (e.g. Farinotti et al., 2017), as well as structural uncertainties in the glacier model itself (e.g. Huss and Hock, 2015). However, several studies have shown that climate model uncertainty largely dominates over glacier model uncertainty (e.g. Marzeion et al., 2018, 2020). We therefore refrain from propagating

uncertainties from subglacial topography and the glacier model through all climate scenarios.

### 3.4.4 Uncertainty in sedimentation rates



Estimating the uncertainty in sedimentation rates is highly challenging as only few data are available for direct benchmarking. We account for uncertainties in our first-order estimates in the same way as for the year of lake formation, i.e. by determining the standard deviation of the results provided by the 13 GCMs considered for each SSP. We deem it likely that actual uncertainties are larger, especially when considering that (i) the underlying processes are only described by highly simplified parameterizations, and (ii) systematic uncertainties due to the chosen model parameters come into play as well. We argue that our simplified approach is defensible as our present understanding does not allow for providing a more complete uncertainty assessment.

## 4 Results

### 4.1 Potential future lakes

In total, we detected 683 potential lakes with an area larger than 5,000 $m^2$ and a maximum depth of at least 5 m (3,600 potential lakes are detected without imposing any criteria on size). They extend over a total area of 45.2 ± 9.3 $km^2$ (Fig. 2A), which is about 4.7% of the presently glacierized area in Switzerland. The total potential lake volume, i.e. the total volume of all detected potential lakes, is 1.16 [1.05, 1.32] $km^3$ (Fig. 2B; numbers provide the mean estimate together with a 95% confidence interval). Whilst the average depth is of only 25.7 [22.2, 29.2] m (Fig. 2C), the maximal depth reaches as much as ~330 m. This extraordinary depth is detected for the largest potential lake, which is situated below Konkordiaplatz, i.e. at the confluence zone of the individual branches of Grosser Aletschgletscher (Fig. 1B). The average elevation of the detected potential lakes is 2776 m a.s.l., although individual lakes are found as high up as ~4300 m a.s.l. (Fig. 2D), below Colle Gnifetti, Gornergletscher. The largest fraction of the identified, potential total volume is contained within a reasonably small number of large depressions: the 11 largest potential lakes, for example, contribute a total volume of 0.54 [0.45, 0.67] $km^3$, corresponding to 46% of the total (Fig. 2B).

To obtain a regional picture, we aggregated the potential lakes among the four main river catchments of Switzerland (Fig. 1). The region with the highest number of potential glacier lakes (447) is the Rhone basin, where their volume corresponds to 2.2% of the total glacier volume at present. In the Rhine and Inn catchments, 185 and 26 potential lakes are found, respectively. The total lake volume in these catchments corresponds to 1.5% (Rhine) and 1.4% (Inn) of the present-day glacier volume. The region with the lowest number of potential glacier lakes (24) is the Po catchment, where the total lake volume is equivalent to 0.6% of today's glacier volume.

In general, the size of the potential lakes is related to the size of the glacier, meaning that the largest potential lakes are also found beneath the largest glaciers. Our results indicate that the four glaciers with the highest potential lake volume might give rise to up to 117 new lakes, accounting for 53% of the anticipated total volume in Switzerland. The five largest potential lakes of the Swiss Alps are characterised in Table 1. To first instance, the correlation between lake size and





glacier extent can be attributed by the fact that larger glaciers have more erosive power, and thus can produce larger overdeepenings (Iken & Bindschadler, 1986).

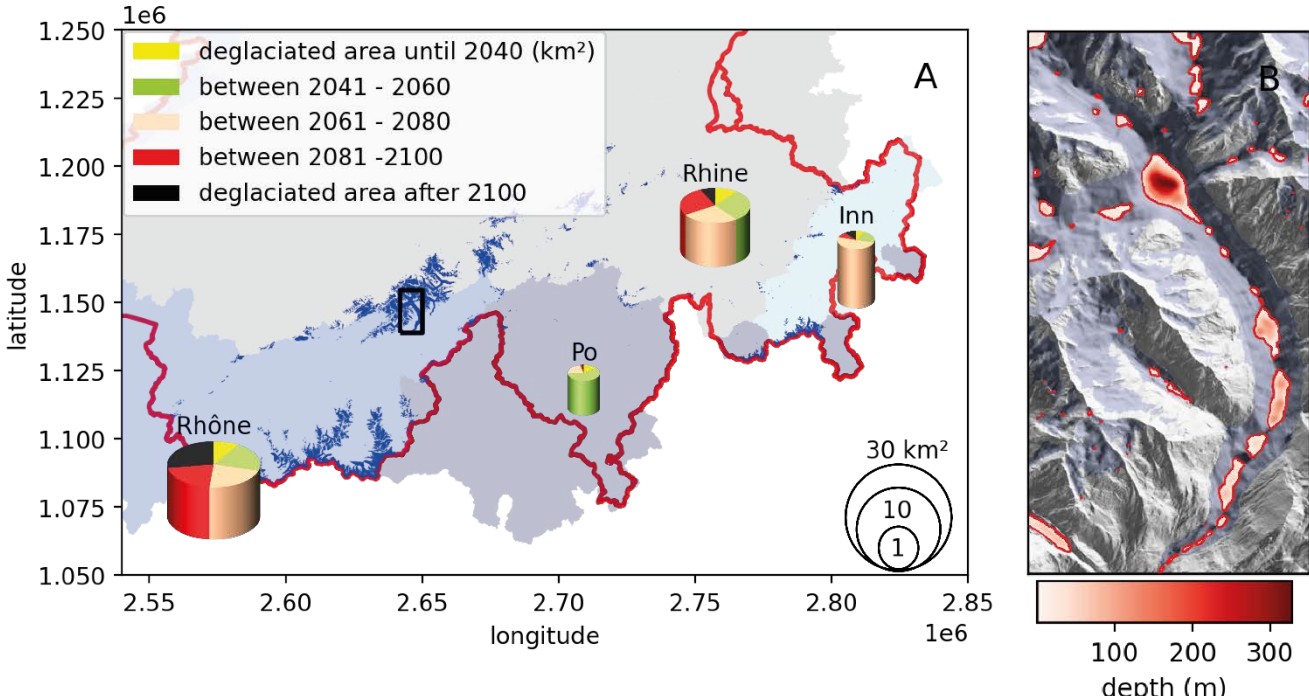


**Figure 1 (A)** Map of Switzerland showing the four main river basins (background colours) used for aggregating the results. Glacier extents according to the SGI2016 (Linsbauer et al., 2021) are shown in dark blue. For each catchment, the temporal evolution of the total area of the detected potential glacier lakes is indicated (pie chart area on logarithmic scale, legend at bottom right). The results are given in 20-year intervals (see legend at top left) and correspond to the simulations driven by the median scenario SSP245. Sedimentation is neglected in this graph. The height of the pies indicates the mean depth of the potential lakes within a given catchment. The black box shows the area enlarged in panel **B**. **(B)** Map of the lower part of Grosser Aletschgletscher, illustrating the morphology of potential glacier lakes. Lakes are contoured in red with depths given by the reddish colours. The SGI 2016 glacier extent (blueish background) is given together with a relief of the swissALTI3D surface DEM and the subglacial topography (grey shading).



Earth **Surface**
**Dynamics**
Discussions

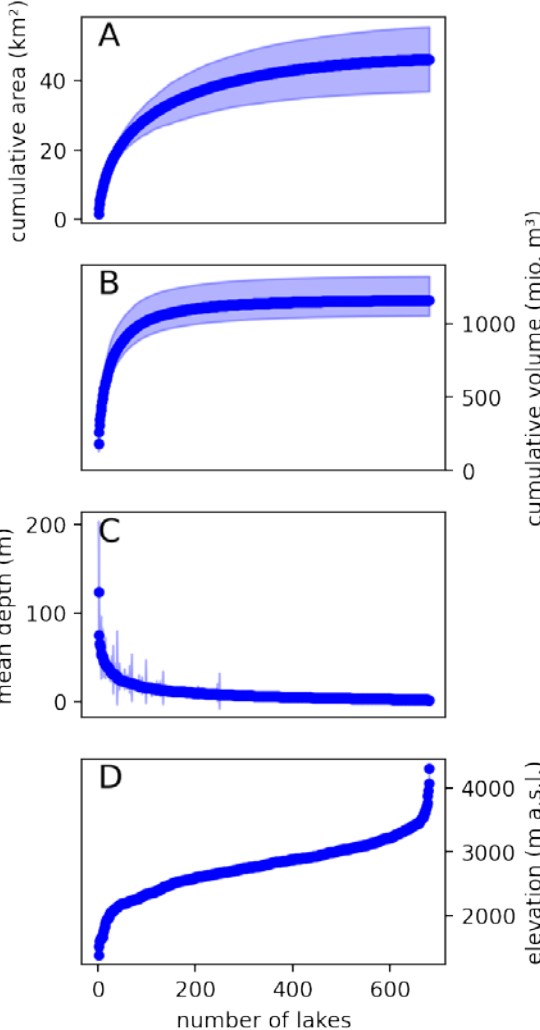


**Figure 2** Cumulative area **(A)** and volume **(B)** of all potential lakes identified within the perimeter of the SGI2016 (Linsbauer et al., 2021). The cumulative uncertainty is given by the shaded bands. Distribution of **(C)** average depth and **(D)** elevation of the maximal lake level of all potential lakes. In panel **C**, shaded bars show the uncertainties of each individual lakes whilst in panel **D** the uncertainty is too small to be seen (<80 m in all cases).






**Table 1** Overview of the five largest potential lakes. The lake volume expected to be ice free until 2050 and 2100 (neglecting sedimentation) is given for SSP245 and is expressed in % of the potential total lake volume. Square brackets provide a confidence intervals for the given quantities (see Sec. 3.4.2). Note that two of the five potential lakes are anticipated for Grosser Aletschgletscher.

| Glacier | Lake-ID | Area (km²) | Elevation (m a.s.l.) | Pot. volume (mio. m³) | Volume free until 2050 (%) | Volume free until 2100 (%) |
|---|---|---|---|---|---|---|
| Gr. Aletschgletscher | B36-26-01 | $1.5 \pm 0.1$ | 2206 | 183 [131, 263] | 0 [0, 0] | 2 [0, 7] |
| Rhonegletscher | B43-03-01 | $1.1 \pm 0.1$ | 2566 | 82 [53, 137] | 0 [0, 0] | 74 [4, 100] |
| Gornergletscher | B56-07-01 | $1.7 \pm 0.1$ | 2187 | 46 [15, 98] | 34 [3, 72] | 95 [58, 100] |
| Gr. Aletschgletscher | B36-26-02 | $0.6 \pm 0.1$ | 1840 | 36 [18, 65] | 0 [0, 0] | 97 [75, 100] |
| Fieschergletscher | B40-07-01 | $0.7 \pm 0.2$ | 2809 | 32 [16, 61] | 0 [0, 0] | 6 [0, 17] |


**Table 2** Overview of the total potential lake volume anticipated to emerge by the years 2050 and 2100 when accounting for sedimentation (first two columns), the percentage of the total lake volume that might be lost because of sedimentation until 2100 (as % of the potential volume), as well as numbers of potential lakes appearing (because of glacier retreat) and

disappearing (because of sedimentation) until the year 2100 (last two columns).

| Climate Scenario | Pot. volume 2050 (km³) | Pot. volume 2100 (km³) | Sedimented until 2100 (%) | lakes appeared by 2100 | lakes disappeared by 2100 |
|---|---|---|---|---|---|
| SSP119 | 0.09 [0.01, 0.18] | 0.20 [0.04, 0.33] | $25 \pm 13$ | $381 \pm 140$ | $186 \pm 65$ |
| SSP126 | 0.10 [0.03, 0.18] | 0.35 [0.12, 0.49] | $21 \pm 7$ | $473 \pm 79$ | $222 \pm 37$ |
| SSP245 | 0.12 [0.04, 0.18] | 0.56 [0.26, 0.67] | $16 \pm 5$ | $570 \pm 61$ | $260 \pm 28$ |
| SSP370 | 0.13 [0.04, 0.22] | 0.82 [0.41, 1.01] | $13 \pm 3$ | $637 \pm 53$ | $277 \pm 27$ |
| SSP585 | 0.19 [0.05, 0.26] | 0.94 [0.51, 1.04] | $11 \pm 3$ | $655 \pm 40$ | $287 \pm 24$ |



### 4.2 Formation of glacier lakes and sedimentation

Depending on the climate scenario, the total volume of potential glacier lakes that will be exposed until the end of this
century can strongly vary (Table 2). Within the first half of this century, the projected glacier retreat and thus the formation
of new glacier lakes is not particularly sensitive to the chosen climate scenarios (Fig. 3A), in contrast to the situation after
2050. SSP585, for example, shows a rapid formation of new lakes after 2050. This formation only decelerates towards the
end of this century, when most glaciers will have vanished entirely. SSP245, instead, leads to an almost linear increase in
the volume of potential new lakes, with the formation of additional $8.2 \pm 2.5$ mio. $m^3$ every year between 2040 and 2100.
According to SSP126 and SSP119, some glaciers might even re-advance towards the end of this century thus filling some
formerly deglacierized lakes with ice once again. This explains the stabilization of the newly emerging potential lake
volume projected according to SSP119 (Fig. 3A). We find that between ~380 (SSP119) and ~655 (SSP585) new glacier
lakes might form until 2100 (Table 2). This is between ~30% and ~65% of the number of new glacier lakes that have been
formed in Switzerland since the Little Ice Age, i.e. in the past ~170 years (Mölg et al., 2021). When accounting for
sedimentation, we find that between ~185 (SSP119) and ~285 (SSP585) new proglacial lakes completely disappear again
until 2100 due to infill. This corresponds to 49% and 44% of the new lakes, respectively, but relates to small
overdeepenings with a rather limited potential volume.

The rate of formation of new glacier lakes differs within the four main river catchments used to aggregate the results (Fig. 1
and 3B). After 2040, the rate is approximatively constant in all catchments, the highest rates being anticipated for the
Rhone and Rhine catchments (annual increase of potential lake volume of 6.82 and 2.65 mio $m^3$ per year, respectively; Fig.
3B). In the Inn catchment, this rate slows down after 2070 since most glaciers are projected to have vanished by then. In the
Po catchment, the same reason causes the potential glacier lake volume to remain virtually unchanged after ca. 2060, with
only four potential lakes not having formed by then.

According to our results, sedimentation of new glacial lakes is an important process that has the potential to fill especially
smaller bedrock overdeepenings. Depending on the climate scenario, we find that between ~185 and ~285 new lakes (i.e.
slightly less than half of the total) will be completely re-filled with sediments by 2100 (Table 2). The total lake volume lost
through this process varies between 32±13% for SSP119 and 11±2% for SSP585, the strong dependence on the climate
scenario being caused by the different glacier extents. Indeed, climate scenarios anticipating less warming (e.g. SSP119)
imply larger glacierized areas, and thus result in both the formation of fewer glacier lakes and generally higher
sedimentation rates owing to the high erosive action of the glaciers themselves. The model results also indicate that large
potential lakes may only lose a small percentage of their volume through sedimentation, whilst smaller sediment sinks can
be filled within years after their appearance.

Our simple model for catchment erosion and the provision of sediment to new glacier lakes (see Section 3.3) shows
complex spatio-temporal dynamics. At the scale of the entire Swiss Alps, we expect sedimentation input into newly formed





lakes to increase until about 2050 (with a range of ±5 years depending on the scenario) and then to decrease towards the end of the century (Fig. 4A). This is explained by the fast exposure of vast proglacial areas with erodible unconsolidated sediments that are assumed to stabilize with increasing age. At the same time, the decreasing thickness of glaciers reduces their potential for bedrock abrasion, with a corresponding reduction of the long-term sediment yield. Furthermore, sedimentation rates in the newly exposed areas also decrease in the long term as smaller lakes, acting as sediment sinks,

will fill up and not retain sediments any longer. Highest sedimentation rates, of up to $1.9\pm0.5$ mio m$^3$ yr$^{-1}$, are expected for the high-emission scenario SSP585, while rates peak earliest for SSP119 and reach $1.2\pm0.5$ mio m$^3$ yr$^{-1}$. When transiently accounting for sediment input, the number of lakes increases throughout the 21$^{st}$ century for all scenarios (Fig. 4B), but only reaches 40-50% of the potential total (see also Table 2). While intermediate scenarios show a continuous growth in lake number, a stabilization is modelled both for the lowest and highest emission scenarios. The reasons are contrasting:

For SSP585, almost all potential lakes are exposed but especially smaller ones are sediment-filled towards the end of the century. For SSP119 in contrast, glaciers stabilize with climate beyond about 2080 (e.g. Compagno et al., 2021) thus not exposing further lakes. The share of potential new lake volume that is filled with sediment is an important measure to assess the evolution of future lakes. While values range between 11% and 25% for the end of the century, the share of already-exposed lakes that fill with sediments again shows a peak at about 40% around 2030 (independently of the SSP),

and decreases afterwards (Fig. 4C). We explain this by the lagged retreat of glacier termini, which first only exposes relatively small lakes that are readily filled with sediments again. When the rate of retreat accelerates towards 2050, and more and larger potential lakes are exposed, the still rising sediment input (Fig. 4A) is more efficiently stored.

The complexity of the temporal sedimentation dynamics is best illustrated at the scale of an individual new glacier lake: Figure 5 shows the volume evolution of the large lake at the snout of Gornergletscher. In a climate scenario implying high

emissions (SSP585), the lake is expected to first appear around the year 2025, to experience fast deglaciation especially after 2040, and to be completely ice-free by about 2060 (Fig. 5A). Whereas modelled overall basin erosion rates show a increase to about 0.9 mm a$^{-1}$ by 2050 and a subsequent decrease to about 0.6 mm a$^{-1}$ (Fig. 5B), actual sedimentation rates of the lake also show strong short-term variability. This is explained by (partly) intermittent exposure of smaller lakes upstream of the considered lake. Indeed, the former act as efficient sediment sinks, and strongly reduce sediment provision

to the areas downstream. Whilst by 2100 only ca. 5% of the considered lake volume is expected to have re-filled by sediments, sedimentation almost keeps pace with the rate of lake volume growth until 2040, i.e. in the phase in which the lake volume is still limited. This suggests that accounting for sedimentation processes is especially important in the early stages of new glacier lake formation.





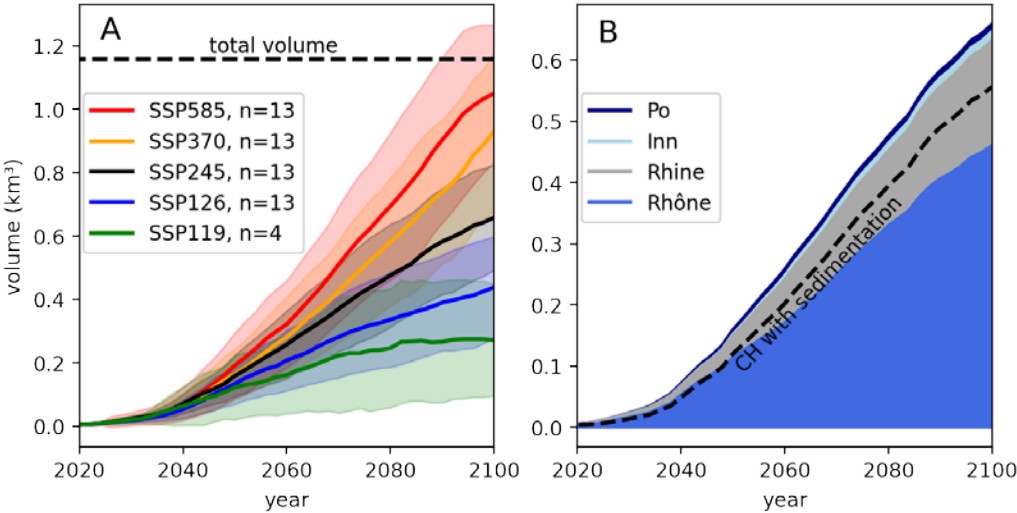


**Figure 3 (A)** Temporal evolution of the total potential glacier lake volume without sedimentation. The total possible volume (i.e. the maximal lake volume that would be realised if all glaciers were to vanish entirely) is indicated with the dashed line (95% confidence interval [1.05, 1.32], not shown). Five different climate scenarios (SSPs) are distinguished. The shaded regions depict the spread between the results obtained with individual climate model chains. **(B)** Temporal

evolution of the total potential glacier lake volume without sedimentation, aggregated for the four main river catchments in Switzerland (see Fig. 1). The results refer to the median scenario SSP245. The dashed line shows the total volume for all four catchments when taking the modelled sediment input into account.

Earth **Surface**
**Dynamics**
Discussions

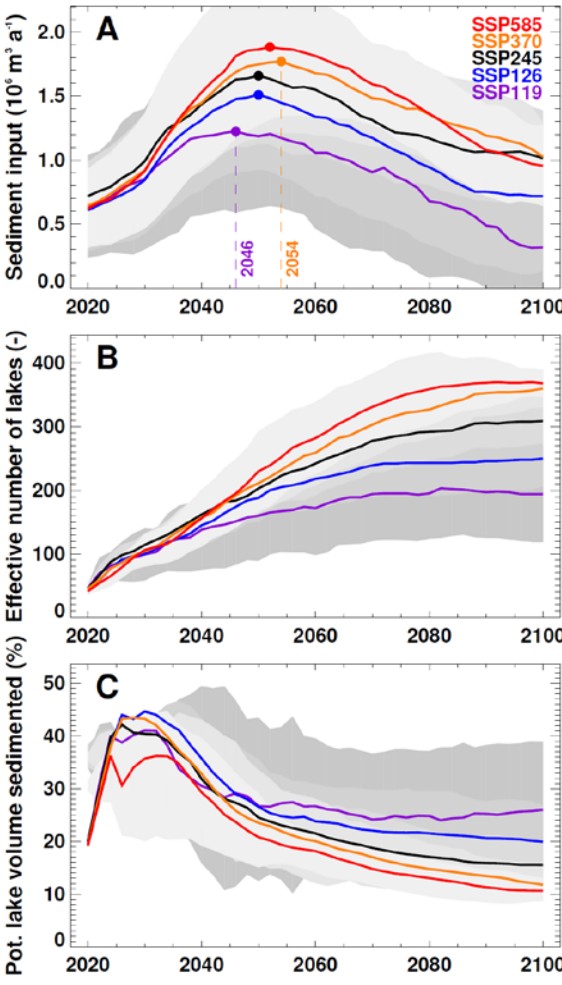

**Figure 4 (A)** Temporal evolution of modelled annual sediment input into new potential lakes formed due to glacier retreat after 2016 (onset of glacier evolution modelling) for all SSPs. Lines are smoothed with a 11-year running mean and represent the average of all 13 GCMs for a given SSP. Dots indicate the timing of the maximum sediment input into lakes. **(B)** Evolution of the effective number of new potential lakes, i.e. accounting for lake basins being completely sediment-filled. **(C)** Share of potentially available lake volume throughout the 21st century being sedimented for the respective scenario.

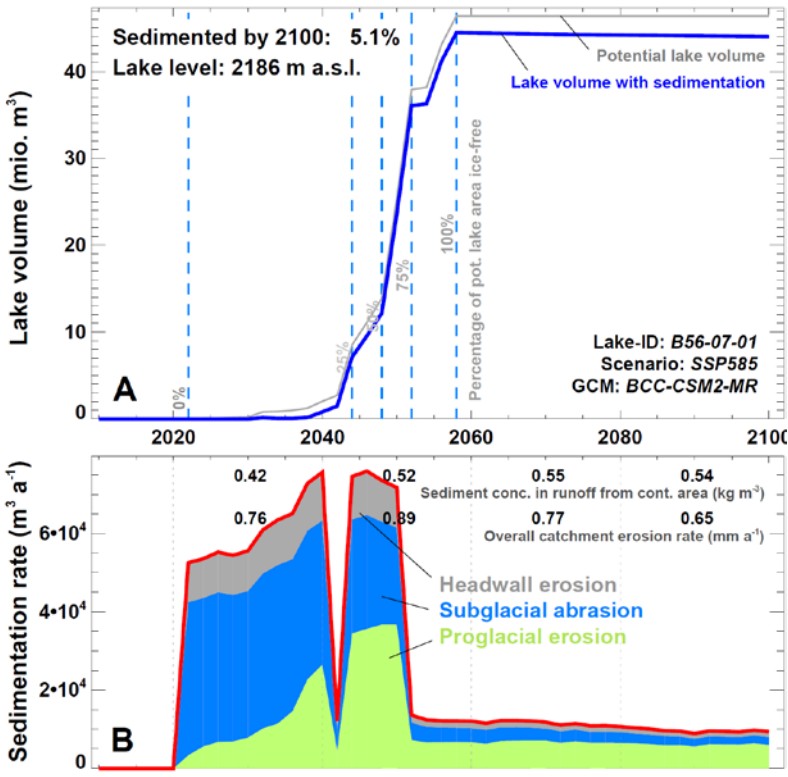

**Figure 5** Temporal dynamics of lake volume growth and modelled sedimentation for the example of Gornergletscher (SGI-ID B56-07), harbouring the third-largest new glacier lake in Switzerland (cf. Table 1). Results refer to a selected GCM amid the climate scenario SSP585. **(A)** Lake volume evolution when neglecting (grey) or accounting for (blue) sedimentation. Percentages of exposed potential maximum lake area over time are shown (grey vertical numbers). **(B)** Modelled lake sedimentation rate, including the partitioning of the three considered sediment sources (cf. Section 3.3). The strong temporal variations are due to the appearance and re-filling of smaller glacier lakes located above the considered lake. Numbers in the upper part of the panel show (i) sediment concentrations in the runoff from the catchment area contributing to the lake at a given point in time, and (ii) modelled catchment erosion rates. The numbers are evaluated over 20-year time steps.



## 5 Discussion

### 5.1 The significance of new glacier lakes

With close to 700 detected overdeepenings, the number of potential new glacier lakes in the Swiss Alps is significant.
Albeit in terms of area and volume a relatively small share of lakes determines the totals (the largest 60 lakes account for
50% of the total area and 80% of the total volume, respectively; Fig. 1), this high number of emerging new lakes is set to
change the appearance of the Alpine landscape – together with glacier retreat, which causes the emergence of these lakes in
first place. A number of impacts can be anticipated from that, including the dynamics of the affected periglacial
ecosystems, the potential anthropogenic use of the newly emerging lakes (e.g. for recreational use, water management
purposes, or hydropower production), or the change in natural hazard potential as significant water masses come to lie in
areas surrounded by steep topography and thus prone to gravitational mass movements such as rockfalls, avalanches, or
even landslides. An in-depth analysis of the possible implications of newly emerging glacier lakes had be conducted in the
frame of the Swiss National Research Programme 61 (NELAK, 2013) and it is beyond the scope of our analysis to
summarize the findings here (for such a summary, refer to Haeberli et al., 2016). What our results show, however, is that
part of this evolution is set to take place in any case, i.e. independently of the climate evolution of the next few decades.
Indeed, as illustrated by Figure 3a for example, the temporal evolution of the emerging new lakes is virtually independent
of the chosen climate scenario until ca. 2050. This is congruent with the anticipated glacier evolution in the Alps (e.g.
Zekollari et al, 2019; Compagno et al., 2021) and is directly related to the glacier's response time (Zekollari et al., 2020),
i.e. the time required by glaciers to adapt their geometry to given climate conditions. In practice, this means that some ~200
potential new lakes are expected to emerge until 2050 in any case (average of all climate scenarios and climate model),
whilst for the end of the century, the number lies between ~240 and ~700, depending on climate scenario and climate
model realization (Table 2).

This large number of potential new lakes calls for a reflection about their future role and embedding. Whilst it might be
anticipated that the large majority of potential new lakes will not trigger major debates and will naturally become part of
the newly emerging landscapes, the situation could be different for some of the larger lakes. Here, the potential for
conflicting interests can be seen, spanning from a strict preservation of their state due to their importance in terms of
emerging natural habitats, to the exploitation for commercial use. Recent debates around this topic – mostly happening in
the context of artificial reservoirs rather than natural lakes (e.g. Kellner, 2019; Kellner and Brunner, 2021; Kellner, 2021) –
have focused on the potentials for multipurpose usage, i.e. the potential of a given lake to satisfy different needs. Such
needs range from the generation of hydroelectricity (e.g. Ehrbar et al., 2018), over the alleviation of water scarcity (e.g.
Brunner et al., 2019), to the management in terms of flood prevention (e.g. Volpi et al., 2018) but are most often centred on
anthropogenic interests. It seems desirable to extend these considerations to aspects that are more difficult to quantify in
terms of economic value, notably including the ecosystem services provided by lakes (e.g. IPBES, 2019) and their role in





the colonization of areas becoming ice free. Whether such considerations will gain momentum remains to be seen, but seem

important in light of the scale of the changes to come.

## 5.2 Actual lake formation and sedimentation

In the presentation of our results, we stressed the wording "*potential* glacier lakes". This is to indicate that even for overdeepenings that are detected through GPR data with confidence, a considerable uncertainty remains about whether a lake will *actually* form. The actual formation can indeed be impeded by the local topography or geology. On the one hand,

narrow outlet channels, which can neither be resolved by the GPR measurements nor be anticipated with the used ice thickness estimation approaches, can be sufficient for preventing a given glacier lake to form at all. On the other hand, overdeepenings eroded into porous or fissured bedrock might never fill with water because of leakage through the underlying rock. In their recent study, Mölg et al. (2021) determined that only about 40% of the area contained within overdeepenings that have been exposed by glacier retreat since the Little Ice Age actually filled with water to form a glacier

lake. If this is assumed to be a characteristic value, it would mean that about $27\pm6$ km$^2$ of the identified potential future lake area ($45\pm9$ km$^2$ in total) will never give rise to an actual lake.

A further process that might hamper actual lake formation is the re-filling with sediment. Our study is the first that attempts to capture this, and does so by using a simple approach that takes into account relevant processes, as well as their spatio-temporal changes. Although more process understanding is needed to increase the reliability of the related results, our

assessment clearly suggests that sedimentation of new proglacial lakes in bedrock overdeepenings can be important, especially for small lakes in the first years after their formation. At the scale of the entire Swiss Alps, our model indicates that – depending on the climate scenario – between 11% and 32% of the potential new lake volume might be filled with sediments by 2100, with higher relative sedimentation for low-emission scenarios. Although other factors, such as the actual volume of overdeepenings and whether they will actually fill with water (see above), are likely more important, it is

clear that sediment input into new proglacial lakes will systematically reduce their volumes. Neglecting this effect thus results in an overestimate for the volume of potential future glacier lakes. Our approach shows complex spatio-temporal dynamics of the sediment input (Fig. 5) and captures effects such as the erodibility of sediments in the upstream catchment, the transport of fine-grained materials by water runoff, as well as the dependence on upstream sediment sinks given by other glacier lakes. Our model has to be understood as a first-order approach to characterize these dynamics, and recognize

the weak validation of our approach. The problem here is that a validation at the scale of the Swiss Alps is almost impossible, given that only few data – mostly referring to individual and small basins – are available. Together with the realization that sediment fluxes from high mountain areas have significantly increased over the past decades (Li et al., 2021), our results might be a motivation to enhance our understanding of the related processes, thus spurring further studies in this domain.





### 5.3 Robustness of detected potential lakes

Despite the unusually comprehensive set of GPR data available to determine the location and size of the potential future lakes, our results are critically dependent on the methods used to infer the subglacial topography in glacier areas that are not covered by the direct measurements. In this respect, we assess the robustness of our estimates by comparing the results obtained when using the two independent methods used by Grab et al. (2021), i.e. GlaTE and ITVEO. When applying the same thresholds for the detection of potential lakes as used for the mean bedrock topography (i.e. maximal lake depth >5 m and lake area >5,000 m$^2$), ITVEO detects 973 lakes and GlaTE 725 (Fig. 5A and B). This is a significantly larger number when compared to the 683 potential lakes detected when using the mean bedrock topography (cf. Sec. 4) and indicates that for the latter, features that are not robust (i.e. features that are not picked up by both GlaTE and ITVEO) are partially removed. This interpretation is supported by (1) the fact that both the total area (Fig. 5A) and the total volume (Fig. 5B) of the potential lakes detected by using the mean bedrock topography is lower than either of the GlaTE or ITVEO results, and (2) visual inspection of the individual results (e.g. Fig. 6).

In general, ITVEO tends to produce more pronounced topographical features whilst the subglacial topography generated with GlaTE is smoother (Fig. 6B). This difference in smoothness results in a higher number of potential lakes being detected with ITVEO than with GlaTE (Fig. 6A) but since the additional lakes are generally small, they contribute little to the total volume of potential lakes (Fig. 5A). With an average depth of 32.2 m, GlaTE also produces slightly deeper lakes than ITVEO (average depth = 26.3 m) whilst the anticipated potential lake area is very similar (Fig. 5B).

The differences between GlaTE and ITVEO is dependent on the available GPR coverage. In particular, the two approaches show larger differences with increasing distance to GPR measurements (Fig. 6B). On glaciers with low GPR coverage, the differences can be large, with some cases showing estimated potential lake volumes differing by up to a factor of two (e.g. Glacier du Trient, SGI-ID B90-02; Glacier de Corbassière, B83-03; Unterer Theodulgletscher, B56-28; Bisgletscher, B58-08; Hüfifirn, A51d-10; or Obers Ischmeer, A54l-31). In general, however, the total volume of potential lakes detected on a glacier-by-glacier basis by GlaTE or ITVEO is very similar, the comparison visualised in Figure 5C showing a correlation coefficient as high as r$^2$=0.96

All of the above considerations indicate that although uncertainties still exist, our approach is reliable and robust in identifying potential glacier lakes that might emerge in future.

Earth **Surface**
**Dynamics**
Discussions

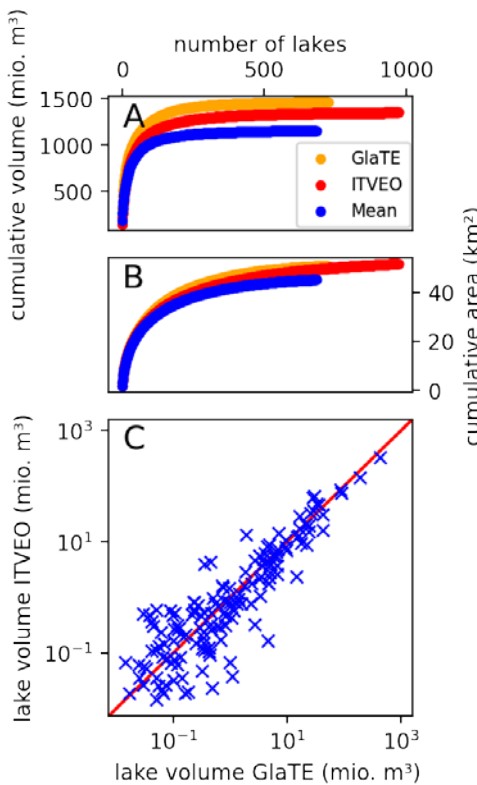

**Figure 5** Cumulative **(A)** volume and **(B)** area of all potential lakes detected with GlaTE (orange), ITVEO (red), and the mean bedrock topography (blue). **(C)** Comparison of the volume of potential lakes per glacier detected with GlaTE and

ITVEO. Along the red line, the volume calculated with the two models is equal. Note the logarithmic scale.

Earth **Surface**
**Dynamics**
Discussions

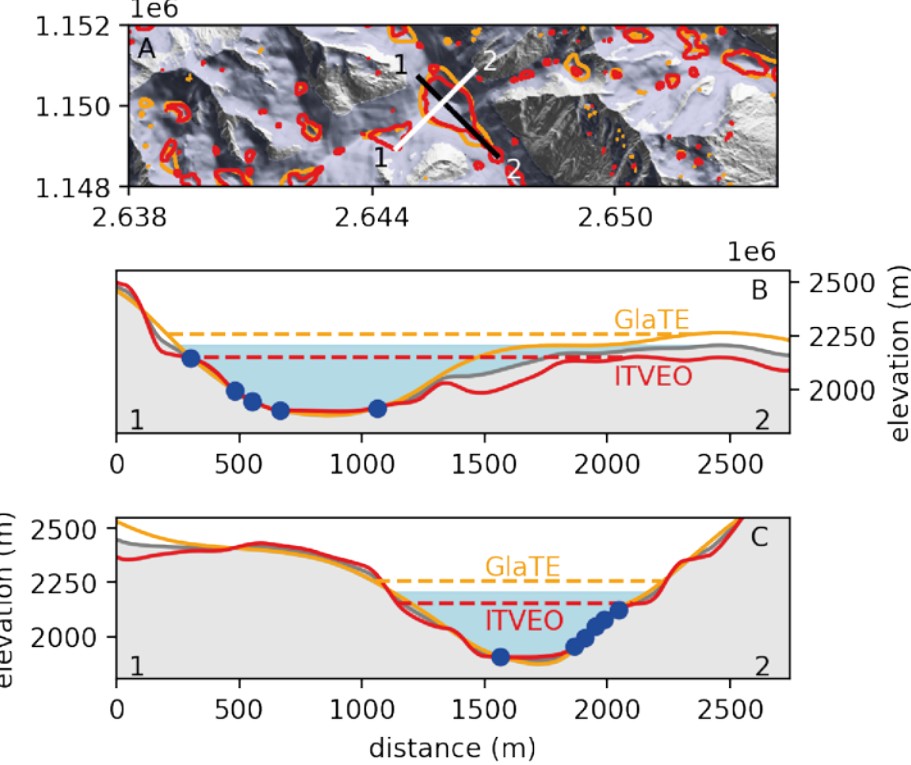

**Figure 6 (A)** Comparison between the potential lakes detected by GlaTE (orange) and ITVEO (red) in the Aletsch region.
The glacier area of the SGI2016 is marked in blue. The black and white lines provide the locations of the profiles displayed
in panels **B** and **C**, respectively. **(B-C)** Illustration of the bedrock topography (orange: GlaTE; red: ITVEO; grey: mean) for
the profiles given in panel **A**. GPR measurements are indicated (blue dots) together with the lake levels resulting from the
three topographies (blue is the lake for the mean topography).

### 5.4 Comparison to previous studies

Our study is not the first one trying to quantify the emergence of future glacier lakes in the Swiss Alps. Linsbauer et al.
(2012) applied the "Glacier bed Topography (GlabTop)" model to estimate the subglacial topography of the glaciers
contained within the Swiss Glacier Inventory 2000 (Paul, 2007) and assumed future lakes to form in the detected bedrock
overdeepenings. By applying a threshold for the lake area of >10,000 m$^2$, they reported that between 394 and 523 new
lakes might eventually form if all glaciers were to disappear, corresponding to a total potential lake volume of between 1.2
and 1.6 km$^3$ (the range in their results reflects two different procedures used to estimate the subglacial topography and the
corresponding overdeepenings). With the same size threshold (i.e. when only considering lakes with an area >10,000 m$^2$),





our results indicate 543 potential lakes for a total volume of 1.15 [1.05, 1.32] km$^3$. Considering that our estimates are based on a comprehensive set of direct ice thickness observations (cf. Sec. 2) whilst the results from Linsbauer et al. (2012) were only based on information of the surface topography, the results are remarkably close. Of note is also the fact that, similarly to our results, the largest potential lake detected by Linsbauer et al. (2012) is situated beneath Konkordiaplatz, at Grosser

Aletschgletscher. Although the potential area of that lake estimated by Linsbauer et al. (2012) is about 1 km$^2$ larger than the area estimated in our study (2.5 km$^2$ compared to 1.5±0.1 km$^2$), the mean depths are in very good agreement (Linsbauer et al., 2012, estimated 100m, whilst our study indicates 123 [88, 178] m). Although this comparison indicates that the total volume of potential future glacier lakes is somewhat smaller than previously estimated, it also shows that the methods used by Linsbauer et al. (2012) were skilful.

More recently, Gharehchahi et al. (2020) used an approach named "Volume and Topography Automation" to estimate the size and distribution of future glacier lakes in the Rhone catchment. They assessed the performance of their approach to be more reliable on glaciers larger than 5 km$^2$, and found 125 potential future lakes with a total volume of 0.68 km$^3$ beneath glaciers of this size class. These numbers are significantly smaller than the 307 potential lakes and the total volume of 0.84 [0.74, 1.01] km$^3$ we determine for the same glaciers. The same applies to the total area of these lakes: whilst Gharehchahi

et al. (2020) computed an area of 19.3 km$^2$, we detected 27.4±4.5 km$^2$. The opposite is true for the average lake depth. The potential lakes identified by Gharehchahi et al. (2020) are slightly deeper (average depth: 34.9 m) than ours (30.5 [26.9, 36.7] m). Considering that – similarly as Linsbauer et al. (2012) –Gharehchahi et al. (2020) only used measured ice thicknesses for model validation but not as direct model input, we suggest that the extensive GPR data available for our study might have been helpful in identifying smaller overdeepenings. Of note is also that the largest lake identified by

Gharehchahi et al. (2020) lies beneath the terminus of Gornergletscher, with a predicted total volume of 96 mio. m$^3$. This is more than twice as much as indicated by our results (46.4 mio. m$^3$). Since the potential area of this lake is similar between the two studies (1.6 vs 1.7±0.1km$^2$) and since the terminal area of Gornergletscher is reasonably well covered by GPR data (see Supplementary Figure S4 in Grab et al., 2021), it is likely that Gharehchahi et al. (2020) overestimated the depth of this lake.


## 6 Conclusions

We quantified the number, area, and volume of the potential new glacier lakes that may form in Switzerland due to ongoing glacier retreat. Relying on the recently released subglacial topography by Grab et al. (2021), we systematically detected subglacial overdeepenings and characterised them in terms of both location and shape. In contrast to previous estimates

(Linsbauer et al., 2012; Gharehchahi et al., 2020), our analysis is based on extensive GPR measurements which significantly increases the robustness of the results. In total, we detected 683 potential glacier lakes with a size larger than 5,000 m$^2$ and a depth exceeding 5 m. Together, they hold the potential of covering an area of 45.2±9.3 km$^2$ and of



containing a water volume of 1.16 [1.05, 1.32] km³. These new lakes are mainly situated in the hydrological basin of the Rhone, which accounts for 70% of the total area and 79% of the total volume of all potential future glacier lakes detected in the Swiss Alps. By relying on results from the glacier evolution model GloGEM (Huss and Hock, 2015), we also assessed the timing by which these future lakes are likely to emerge. We show that by 2050, more than a hundred new glacier lakes might form depending on the climate scenario, potentially giving rise to a total lake volume of up to 0.19 km³. Whilst the mid-term evolution of the future glacier lakes shows little sensitivity to the climate scenario that will actually materialize, it might depend more critically on the local sediment inputs. With a first analysis, we estimate that up to one quarter of the new glacier lake volume might have re-filled with sediments by 2050 – the process being of particular importance for small lakes. The influence of the climate scenario will be much more prominent in the second half of this century. By 2100, climate scenarios implying high emissions suggest that up to 90% of the total volume of potential lakes might have formed, whilst only 23% of the volume might form if emissions were abated quickly, hence, slowing glacier recession. The results reiterate the rapid changes that have to be expected in alpine regions, and highlight that relatively little time is available to adapt to these changes. Whilst glacier lakes can be of high ecological relevance and might be attractive for a number of purposes ranging from water management to recreational use, they also bear some hazard potential – outbursts from glacier lakes being one of the most common and far-reaching hazards related to glaciers (Haeberli et al., 1989; Carrivick and Tweed, 2013). Our results may serve as a base to assess both positive and negative implications of these elements of our future landscapes.

**Data availability**

During review, the results generated within this study are available in digital format at https://drive.switch.ch/index.php/s/MV5hd1HS8AxrueM. This URL will be replaced with a DOI-handle in the event of the paper being accepted for publication.

**Author contributions**

DF and MH conceived the study. TS and RE performed the lake-volume calculations with the help of EH and DF. MH performed the GloGEM simulations and modelled the future sediment inputs. DF designed the figures, which were realized by TS, RE and MH. TS, DF, and MH drafted the manuscript, to which all authors contributed.

**Competing interests**

The contact author has declared that neither they nor their co-authors have any competing interests.



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
