# Peer review of "Volume, evolution and sedimentation of future glacier lakes in Switzerland over the 21st century"

_Earth Surface Dynamics, 2022_

## Author Comment (AC1)

**Author's response to the comments received for manuscript "esurf-2022-12"**

As requested by the handling editor, the following pages contain a point-by-point reply to the comments provided by the referee that reviewed our first submission (esurf-2022-12).

Each of the referee's comment (**RC**) is numbered, and given in *black italics*. If a comment contained several points, we numbered them in *brown*, and address them individually in our author replies (**AR**, blue text). Where appropriate, the revised version of the text appears below in smaller font size and quotation marks (line numbers refer to the revised manuscript-version in which changes are tracked).

We are convinced that the constructive suggestions provided by both referees have improved our work, and thank the referees very much for the time invested in reviewing our submission.

**Referee #1 (Greta Wells):**

**General Comments**

*[RC 1.01] This manuscript thoroughly discusses future glacial lake evolution in the Swiss Alps derived from newly acquired glacier topographic data. It clearly frames the study in the context of previous research, explains existing data gaps, and offers a robust methodology to fill these gaps. The manuscript adequately answers the fifteen evaluation criteria questions, scoring particularly highly on organization, explanation of methods for reproduction by other scientists, and discussing the study in relation to existing work and future research implications.*

*The paper does an excellent job of maintaining the balance between model generalization and case study specifics. One of its strongest features is comprehensively explaining the selection process and potential uncertainties or unknowns for each model parameter. The authors also demonstrate a firm understanding of how field variables (i.e. real-life topography, sediment, and glacier melt dynamics) are represented by model components, and how changes in field variables will influence model input parameters and results/outputs. I think such a clear link is often missing in modeling papers.*

**[AR 1.01]** We very much thank the referee for the time invested in reviewing our submission and are very pleased by the positive appreciation of our work. In particular, we are happy to hear that the study is recognized to hold value also beyond the Swiss Alps, i.e. the specific case study we address, and that the submission provided a balanced impression in its entirety. We would also like to thank the referee for the very constructive comments provided here below, and are happy to say that we shall be able to include the vast majority of them in our revised manuscript.

\-\-\-\-\-\-\-\-\-\-\-

**Specific Comments**

*[RC 1.02] Title: Maybe replace "formation" with "evolution" to imply how lakes will continue to change through time rather than when they will initially form. "Evolution" would also better describe lakes that disappear due to sediment infilling. Also consider adding the study time scale into the title – something like Volume, evolution, and sedimentation of glacier lakes in Switzerland over the 21st century.*

**[AR 1.02]** Both suggestions look very pertinent to us and we would like to adopt them both. The new title will thus read:

**Lines 1-2:** "Volume, evolution, and sedimentation of glacier lakes in Switzerland over the 21$^{st}$ century"

\-\-\-\-\-\-\-\-\-\-\-

*[RC 1.03] Do mass wasting (i.e. from surrounding slopes) and/or supraglacial debris (i.e. transferred from the glacier surface to the lake) significantly contribute to sediment infill? Maybe not, but is it worth considering?.*

**[AR 1.3]** Whether the mentioned processes significantly contribute the sediment infill is difficult to assess, but it is certainly conceivable for these processes to play a role. Conceptually, one could argue that the processes are implicitly accounted for in our method, and in the indices used for subglacial abrasion and headwall erosion in particular. Indeed, by ensuring that on average over all glaciers we obtain sediment production rates that are consistent with long-term observations (cf. our former L215ff), any processes that lead up to these rates are considered – even in the case that the process cannot be named explicitly. We add this consideration in the revised text, as it seems important to us:

**Lines 227-232:** "Albeit empirical, the approach allows for capturing both the high variability in erodibility of different glacier catchments, as well as the temporal dynamics of sediment yield in the transition from glacierized to ice-free basins. Also note that by choosing $c_{sed,in\ ref.}$ to reflect long-term observations, the procedure implicitly accounts for any other processes that might contribute to the generation and evacuation of sediments in the considered catchments – such as mass wasting from surrounding slopes, or the advection of supraglacial debris trough ice flow, for instance."
* * *
*[RC 1.04] While the paper explains the model generalizations and parameter uncertainties for lake bathymetry, I think this point merits further discussion. For example, are variabilities in overdeepening morphologies expected at individual glaciers or between the four river catchments in the study area? Even though a thorough analysis of basin morphometry is beyond the scope of this paper, specific discussion in the context of the study area would be interesting.*

**[AR 1.04]** We are not entirely sure to understand this comment, also because we are unsure to which part of the manuscript it is referring to. We agree with the reviewer that a morphometric analysis of the potential future lake basins is beyond the scope of our study. We also note that such an analysis (i.e. the morphometry of the basins itself) would be the direct result of the study by Grab et al. (2021) and thus not a new development. We will nevertheless try to capture at least part of this consideration by amending Section 5.2 as follows:

**Lines 489-497:** "A further process that might hamper actual lake formation is the re-filling with sediment. Although the literature addressing the interplay between sediment dynamics and bedrock overdeepenings is vast (e.g. Hooke, 1991; Alley et al., 2003; Cook and Swift, 2012; Swift et al., 2021), our study is the first that attempts to capture this process in the context of the formation of potential new glacier lakes. It does so by using a simple approach that takes into account relevant processes, as well as their spatio-temporal changes. Although more process understanding is needed to increase the reliability of the results, and although our approach has not been evaluated in combination with the morphometry of the potential new lake basins, our assessment clearly suggests that sedimentation of new proglacial lakes in bedrock overdeepenings can be important – especially for small lakes in the first years after their formation. At the scale of the entire Swiss Alps, our model indicates that […]"
* * *
*[RC 1.05] Similarly for sediment infill – the paper does a great job of explaining the model parameters/uncertainties for sedimentation rate, but I think it's worth expanding the discussion on how local/site-specific variations may influence results. For example, (1) are there significant anticipated differences in sedimentation rate (i.e. due to local bedrock lithology or erodibility) between the four studied Alpine catchments? (2) Also, in section 5.2 (lines 480-484), it would be helpful to detail the type of field validation required (i.e. bathymetric surveys or sediment influx river/lake measurements).*

**[AR 1.05]** Both are valid points, and we tried to address them at least in part. For (1), we are unfortunately unable to provide an answer, since our approach does not take bedrock lithology explicitly into account. While this would certainly possible in principle (especially for a study region such as Switzerland, in which extensive geological information is available), such a procedure would hinder the applicability of our approach, which so far is only based on morphological information that can be retrieved from basic glacier characteristics and our modelling results. We also argue that investigating the effects of lithology on glacial erosion would require a dedicated study. Indeed, our approach is unable to fully resolve local erosion processes, and additional work and data would thus be necessary. We will aim to express this thought in the Conclusions, also in response to **RC1.08**, further below. More specifically, we will add the following paragraph:

**Lines 613-621:** "In terms of methodology, our analysis is strongly based on the unique dataset that is available for the subglacial topography of the Swiss Alps (Grab et al., 2021). This notwithstanding, our approach for estimating the future evolution of potential glacier lakes is transferrable, at least in principle, to other glacierized regions on Earth too. This is particularly true also for our first-order analysis of future lake sedimentation rates – as long as average sediment production rates can be estimated for the region of interest. Indeed, our results are based on a combination of modelling results that are available at the global scale (Huss and Hock, 2015) and on morphological characteristics that can be retrieved from surface characteristics. Albeit such an approach entails some important simplifications – such as neglecting the influence of bedrock lithology in the estimated sedimentation rates, for instance – our results may serve as a base for future studies. Since glacier lakes are set to become an important element of our future landscapes, we argue that such studies can be of high relevance."

For (2), we will add a hint for which type of field measurements would be necessary to validate our approach in detail, and for why these do not exist to date:

**Lines 507-511:** "A rigorous validation would not only require long-term time series of catchment-wide erosion rates for a representative set of glaciers, but also a partitioning of the individual sediment sources within the individual basins. Such information is extremely rare, if not inexistent, as it is very difficult to devise measurements that would enable such a partitioning in first place – let alone maintaining them over a period of time that would allow for establishing a relationship between a change in partitioning and glacier retreat."
* * *
**[RC 1.06]** *The figures are great, but I think the paper would be strengthened by adding a schematic diagram to accompany the sediment input graph in Figure 5. Maybe a cartoon visualization of a lake (like the cross-section shown in Figure 6) that shows the locations on the landscape of sediment source zones (headwall erosion, subglacial abrasion, and proglacial erosion). Also maybe draw upstream lakes to illustrate sediment trapping. Though the information is described in the text and existing figures, a "landscape view" would help to visualize sediment sources and transport.*

**[AR 1.06]** This is an excellent suggestion, and we have produced such a cartoon now (reproduced here below, on the next page). We intend to include this cartoon as panel "C" in Figure 5. The Figure's caption will be changed accordingly.

[Figure]

**Figure 5:** "Temporal dynamics of lake volume growth and modelled sedimentation for the example of Gornergletscher (SGI-ID B56-07; see Figure 1 for location), harbouring the third-largest new glacier lake in Switzerland (cf. Table 1). Results refer to a selected GCM amid the climate scenario SSP585. (**A**) Lake volume evolution when neglecting (grey) or accounting for (blue) sedimentation. Percentages of exposed potential maximum lake area over time are shown (grey vertical numbers). (**B**) Modelled lake sedimentation rate, including the partitioning of the three considered sediment sources (cf. Section 3.3). The strong temporal variations are due to the appearance and re-filling of smaller glacier lakes located above the considered lake. The red circled numbers represent distinct phases of sediment dynamics as illustrated in panel **C**. Values in the upper part of the panel show (i) sediment concentrations in the runoff from the catchment area contributing to the lake at a given point in time, and (ii) modelled catchment erosion rates. Both quantities are evaluated over 20-year time steps, and the latter quantity is obtained by assuming an average density for eroded materials of 2800 kg m$^{-3}$ and by dividing the catchment-wide sediment yield (now expressed as m$^3$ yr$^{-1}$) by the catchment area. (**C**) Illustration of the individual phases of sediment dynamics given in panel B. Further explanations are given in the panel's bottom right corner. The illustration is adapted from Otto (2019)."
* * *
*[RC 1.07] A map showing the locations of the glaciers mentioned in the text would be helpful (i.e. those in Table 1) – though I did notice that the lake polygon shapefiles are online in "data availability".*

**[AR 1.07]** Also this is a very good suggestion. To take it into account, we will amend Figure 1A and include the names of the glaciers mentioned in Table 1. Similarly, we will include the location of the glacier shown in Figure 5. We think that this solution (i.e. the inclusion of the names in Fig. 1) is more elegant than adding a potential, separate map. The amended Figure is shown here below (next page), and takes into account **RC 1.23** and **RC 2.10** too. The Figure's caption was be adapted to flag the indicated locations explicitly.

[Figure]

**Figure 1:** (**A**) Map of Switzerland (see inset at bottom left for location within Europe) showing the four main river basins (background colours) used for aggregating the results. Glacier extents according to the SGI2016 (Linsbauer et al., 2021) are shown in dark blue. For each catchment, the temporal evolution of the total area of the detected potential glacier lakes is indicated (pie chart area on logarithmic scale, see legend at top left). The results are given in 20-year intervals and correspond to the simulations driven by the median scenario SSP245. Sedimentation is neglected in this graph. The height of the pies indicates the mean depth of the potential lakes within a given catchment. Map coordinates are given in the so-called CH1903+ / LV95 system (official Swiss coordinate system). Glaciers that appear in Table 1 or Figure 5 are labelled ("gl." is the abbreviation for "gletscher"). The red box shows the area enlarged in panel **B**. (**B**) Map of the lower part of Grosser Aletschgletscher, illustrating the morphology of potential glacier lakes. Lakes are contoured in red with depths given by the reddish colours. The SGI 2016 glacier extent (blueish background) is given together with a relief of the swissALTI3D surface DEM and the subglacial topography (grey shading). The red box indicates the area shown in Figure 7.
* * *
*[RC 1.08] Conclusion (section 6): It would be nice to expand discussion of future climate change (lines 575-579) in terms of: (1) Specific glacial lakes in the Swiss Alps. Based on results, do certain river catchments/glacial lakes in the study area have higher outburst flood risk, greater hydropower potential, or higher ecological relevance? (2) Other glacial regions worldwide. How can this method be applied to other regions – i.e. are there any factors that make its application unique to the Swiss Alps? Even if this is beyond the scope of the paper, I think it is useful to develop this idea in a few more specific sentences.*

**[AR 1.08]** (1) Albeit this is certainly a valid suggestion, it would be difficult for us to make reliable claims about changes in flood risk, hydropower potential, or ecological relevance. Indeed, we argue that for being of value, such claims would need to be based on dedicated data and a corresponding analysis, which clearly extends beyond the scope of the study. In particular, we fear that only adding a rough, qualitative assessment of such changes could bear some pitfalls in terms of communication: we can only speculate, for example, about how policymakers and stakeholders of a given valley would react if our discussion was to state that their valley is likely to hold a larger flood risk than previously thought. We thus suggest to only slightly expand the discussion in order to point out that our results now provide the basis for conducting such type of follow-up analysis:

**Lines 604-612:** "Whilst glacier lakes can be of high ecological relevance and might be attractive for a number of purposes ranging from water management to recreational use, they also bear some hazard

potential – notably including potential outbursts upon failure of their embankments, or flood waves triggered by mass movements stemming from the steep slopes that often surround Alpine glacier lakes (Haeberli et al., 1989; Carrivick and Tweed, 2013; Haeberli et al., 2017; Haeberli and Drenkhan, 2022). A quantification of the changes in such processes remains beyond the scope and the capabilities of our study, but we suggest that the related consequences should be at the centre of attention in follow-up impact assessments."

(2) This suggestion is very valid too and we would like to take it on board by extending the conclusions as detailed in our **AR1.05**.
* * *
**Technical Corrections**

**[RC 1.09]** *Line 31: "…amongst other features" (or add a similar word).*

**[AR 1.09]** We will reword as suggested.
* * *
**[RC 1.10]** *Line 57: Switch order: "205 additional lakes…"*

**[AR 1.10]** The order will be switched.
* * *
**[RC 1.11]** *Line 62: replace "or" with "and".*

**[AR 1.11]** The named regions (High Mountain Asia, the Andes, or the European Alps) are examples for where such studies exist but the list is not exhaustive. We feel that using "and" would suggest such exhaustiveness, which we find misleading. We would thus keep the sentence unchanged.
* * *
**[RC 1.12]** *Line 68: "…once the glacier has retreated…".*

**[AR 1.12]** We will reword as suggested.
* * *
**[RC 1.13]** *Line 85: replace "combing" with "combining".*

**[AR 1.13]** Sure. We apologise for this typo.
* * *
**[RC 1.14]** *Line 111: "… has been shown to yield the most robust results"*

**[AR 1.14]** We will reword as suggested.
* * *
**[RC 1.15]** *Line 114: remove "to" : "…as the "mean bedrock topography"".*

**[AR 1.15]** We will reword as suggested.
* * *
**[RC 1.16]** *Line 134: remove "as" : "…the procedure described above."*

**[AR 1.16]** This was a mistake from our side. "As" will be removed.
* * *
**[RC 1.17]** *Line 157: maybe replace "political environment" with "policy decisions"?*

**[AR 1.17]** Yes, that indeed sounds like the better wording. We thank for the suggestion and will amend correspondingly.
* * *
**[RC 1.18]** *Line 174: change to "after glacier retreat" (remove "-ed")*

**[AR 1.18]** We will reword as suggested.
* * *
**[RC 1.19]** *Line 189: I think bedrock lithology should be included as a factor in basin erodibility.*

**[AR 1.19]** We absolutely agree and will add this wording as suggested.
* * *
**[RC 1.20]** *Line 239: "The same is true when comparing the lakes…"*

**[AR 1.20]** We will reword as suggested.
* * *
**[RC 1.21]** *Line 250: "The uncertainty in lake volume is also controlled by…"*

**[AR 1.12]** We will reword as suggested.
* * *
**[RC 1.22]** *Line 302-303: "In the first instance, the correlation between lake size and glacier extent can be attributed to the fact that…"*

**[AR 1.22]** We will reword as suggested.
* * *
**[RC 1.23]** *Line 305 (Figure 1): (1) delineate the catchment boundaries more clearly on the map – it is difficult to distinguish between the Rhine and Inn basins, particularly. (2) It would also be helpful to add an inset map showing the study area location within the larger region/Switzerland. (3) The label "B" also does not clearly show up on the panel.*

**[AR 1.23]** All three suggestions are very pertinent, and we have amended the figure correspondingly. The new figure, as we will include it in the revised version of the manuscript, is shown in **AR 1.07**.
* * *
**[RC 1.24]** *Line 315 (Figure 2): add a temporal reference frame (what time scale does this show?)*

**[AR 1.24]** The individual panels refer to the glacier-free topography, i.e. to the hypothetical situation in which all glaciers would disappear entirely. None of the considered climate scenarios imply such a disappearance within the 21st century (even for the very warm scenario SSP585 some high-elevation parts remain glacierized) and thus there is no real time frame associated to this figure. We will amend the caption to clarify this:

**Fig. 2, caption:** "[…] Note that all panels refer to the ice-free topography, i.e. to the hypothetical situation in which all glaciers would melt entirely."
* * *
*[RC 1.25] Line 319: remove "s" in lakes (…of each individual lake)*

**[AR 1.25]** "s" will be removed.
* * *
*[RC 1.26] Line 323: remove "a" (…provide confidence intervals…)*

**[AR 1.26]** "a" will be removed.
* * *
*[RC 1.27] Line 346: …disappear again by 2100?*

**[AR 1.27]** We will reword as suggested.
* * *
*[RC 1.28] Line 349: …the rate is approximately constant…*

**[AR 1.28]** We will reword as suggested.
* * *
*[RC 1.29] Line 395 (Figure 3): perhaps this is obvious and I missed it, but explain abbreviation "CH" on panel B.*

**[AR 1.29]** We apologise. "CH" is an abbreviation for "Switzerland" but was not clear in the context. We will replace the label with "Total lake volume accounting for sedimentation", which should avoid any confusion.
* * *
*[RC 1.30] Line 403 (Figure 4): explain what the grey areas denote – I assume it's uncertainty, but clarify it in the caption.*

**[AR 1.30]** Correct, the grey bands represent an uncertainty stemming from the variability in various climate models run under a given SSP. The caption will be amended to clarify that:

**Fig. 4, caption:** "[…] In all panels, the grey bands denote the variability stemming from the individual climate models contained within a given SSP."
* * *
*[RC 1.31] Line 425: "Although in terms of area and volume…"*

**[AR 1.31]** We will reword as suggested.
* * *
*[RC 1.32] Lines 427-8: "…in the first place…"*

**[AR 1.32]** We will reword as suggested.
* * *
*[RC 1.33] Line 432: …glacier lakes has been conducted…"*

**[AR 1.33]** We will reword as suggested.
* * *
*[RC 1.34] Line 443: perhaps this is a technical term I'm not familiar with, but the word "embedding" is unclear in this sentence. Maybe replace with "position" or "formation"?*

**[AR 1.34]** We will reword with "implications", which seems the more general term to us.
* * *
*[RC 1.35] Line 454: I think you mean plant/vegetation/floral colonization, but maybe add a word to clarify this.*

**[AR 1.35]** We will specify by explicitly stating "colonization of areas becoming ice free *by plants and other living beings*".
* * *
*[RC 1.36] Line 469: use another word besides "related"—maybe "presented"?*

**[AR 1.36]** We believe that the easiest solution is to remove the word altogether. The sentence will then read *"[…] to increase the reliability of the related results"*.
* * *
*[RC 1.37] Line 503: "The differences between GlaTE and ITVEO are dependent…"*

**[AR 1.37]** We will correct as indicated.
* * *
*[RC 1.38] Line 516 (Figure 6): add an inset map to show the location of this site within the study area/Swiss Alps.*

**[AR 1.38]** See AR1.07: the information about the location of individual glaciers will be included in Figure 1. We will extend the caption of Figure 6 (actually, we noted that this should be called Figure 7, and not 6), to clarify that:

**Fig. 7, caption:** "[…] Comparison between the potential lakes detected by GlaTE (orange) and ITVEO (red) for Gr. Aletschgletscher (see Fig. 1 for location). […]"
* * *
*[RC 1.39] Line 568: specify when "mid-term" is (mid-century?)*

**[AR 1.39]** We will specify by saying "Whilst until mid-century the evolution of the future glacier lakes shows little sensitivity to […]".

---

## Author Comment (AC2)

**Author's response to the comments received for manuscript "esurf-2022-12"**

As requested by the handling editor, the following pages contain a point-by-point reply to the comments provided by the referee that reviewed our first submission (esurf-2022-12).

Each of the referee's comment (**RC**) is numbered, and given in *black italics*. If a comment contained several points, we numbered them in *brown*, and address them individually in our author replies (**AR**, blue text). Where appropriate, the revised version of the text appears below in smaller font size and quotation marks (line numbers refer to the revised manuscript-version in which changes are tracked).

We are convinced that the constructive suggestions provided by both referees have improved our work, and thank the referees very much for the time invested in reviewing our submission.

**Referee #2 (Jan-Christoph Otto):**

**General Comment**

*[RC 2.01] The authors present a study on the future evolution of glacial lakes beneath glaciers in Switzerland. While this task has been previously performed for the same study area using a comparable approach, this study adds three significant new aspects to the procedure. The authors use an ice thickness model that is based on a large data set of GPR measurements for many glaciers (1). Even though the spatial assessment of the ice thickness is still based on models, the models used here have the potential to be more close to reality compared to the previous approaches that where based on glacier surface topographies only. Furthermore (2), the approach uses an established model to simulate the release of the potential overdeepenings by modelling glacier volume changes in relation to climate change for different climate scenarios. Finally, (3) the study for the first time accounts for the potential refilling of the exposed overdeepenings by generating a time- and space-dependent approximation of the sedimentation rate at various future stages of catchment and glacier evolution. This approach tackles the highly relevant uncertainty of the true lake evolution and potentially generates a more realistic picture of potential future lakes, despite other sources of uncertainty.*

*The study is well laid out and the manuscript has been produced with great diligence and logic. The methods applied uses data and approaches based on various previous studies published in the recent past. Therefore, methods description is focusing on references to existing papers. Solely the approach to quantify the sediment infill rate adds a new methodological step in this study. This approach is clearly presented, even though some issues arise (see below). However, the procedure presented is convincing and represent a logical way of assessing this critical parameter of lake sedimentation, where very little data is available so far. All results are clearly presented and visualized at good quality. The discussion states the relevant and critical aspects and implications of the approach and topic in general. The authors compare their results to two previous similar studies considering a good agreement with the approach by Linsbauer et al. (2012) and larger discrepancies to the other previous study.*

*I consider the manuscript a valuable contribution to the issue of future evolution of glacial lakes. Especially the accounting for sediment refill adds an important new dimension and the results in relation to future glacier and sediment dynamics present highly valuable new insights into the future of glacial and proglacial sedimentary systems, despite the rather simple approximation of glacial erosion and lake sedimentation. It therefore represents a significant improvement compared to previous studies and is worth publishing. I have only few comments and minor issues to consider.*

[AR 2.01] We thank the referee very much for both the time invested in reviewing our manuscript and the very positive assessment of our work. We are particularly pleased to read that the innovative aspects of our work are recognisable, that the contribution is deemed to be of relevance, and that our methods seem to have been presented in a clear manner. We are convinced that the constructive comments provided by the referee in the following have further strengthened our work and would like to thank the referee for that.
* * *
**Specific Comment**

*[RC 2.02] Section 3.3. – I have some concerns with the use of the variable $\alpha_{crit}$ in the estimation of the $Sed_{in}$ components. For (1) abrasion, the variable makes sense as is. For (2), increase in deglaciarized area, and (3), glacial and periglacial erosion, I would suggest to reconsider your approach or the description of it. From my understanding $\alpha_{crit}$ represents the mean slope of all glaciers of the SGI2016 (L196 "$\alpha_{crit}$ are critical values for mean thickness and slope that correspond to an average Swiss glacier"). For parameter (1), it makes sense*

*to me to use an overall mean for all glacier in the equation. Here you compare h and slope of individual glaciers with overall means across the SGI2016 dataset to generate an index of abrasion, which differs between glaciers due to size and topography. However, for parameter (2) and (3) I think it would make more sense to use $α_{crit}$ as the mean slope of the individual glacier and not the overall mean. Since all other terms of the equation are referring to the individual glacier, I don't understand why slope does not. Maybe it's just a mistake in describing the equations. Please reconsider this issue.*

**[AR 2.02]** We perfectly understand the referee's concern and realize that we did not manage to properly convey our reasoning behind the individual terms featuring in the indices presented with the equations of Section 3.3. The main idea is to capture variations in sediment production rates ($Sed_{in}$ in the referee's wording) by accounting for the morphological characteristics of individual glaciers and their surroundings. This is done by (1) ensuring that, on average over all glaciers, $Sed_{in}$ matches an average sediment production rate as determined from literature values (former Lines 215-217), and (2) scaling this average rate on a glacier-by-glacier basis (this scaling is performed through the individual components of the indices) depending on how the morphology of a specific glacier compares to the average morphology of all glaciers. In this sense, $α_{crit}$ is indeed meant to describes the average slope of all glaciers, and the terms $α_{proglacial}/α_{crit}$ (for proglacial erosion in Eq. 3) and $α_{headwall}/α_{crit}$ (for headwall erosion in Eq. 4) are meant to modulate the glacier's sediment production rate depending on the average slope of their specific proglacial area ($α_{proglacial}$) or headwall ($α_{headwall}$). What we realized thanks to the referee's comment, is that we failed to state that the $α_{crit}$ values of the three parametrized processes (i.e. subglacial abrasion, proglacial erosion, and headwall erosion) can be different. We will both clarify this difference by explicitly introducing three values ($α_{crit,1}$ , $α_{crit,2}$ , and $α_{crit,3}$ ) and better clarify the reasoning behind our procedure by adding the following information to the text:

**Lines 194-198:** "[…] The main idea behind our approach is to use a set of three dimensionless indices for scaling the rate of sediment input to each lake with the morphological characteristic of a given glacier. This scaling is thereby controlled by how a given morphological characteristic – such as the glacier surface slope or the size of the glacier proglacial area, for instance – compares to the average characteristic of all glaciers in the sample. […]"

**Lines 207-209:** "[…] $α_{crit,2}$ is the average proglacial slope of all Swiss glaciers, and $A_{basin}$ is the total area of the basin. Similarly to $i_{abrasion}$, the index is meant to capture relative variations between glaciers that are caused by differences in morphology, and indeed, a large, steep and recently deglacierized proglacial area will result in higher $c_{sed,in}$ than a small, flat, or long-established one. […]"

**Lines 217-119:** "[…] Similarly as for $i_{abrasion}$ and $i_{proglacial}$, the ratio $α_{headwall}/α_{crit,3}$ is meant to capture relative variations between glaciers, $α_{crit,3}$ being the average headwall slope of all glaciers in the sample. The square for this equation term intends to qualitatively capture the exponential effect that […]"
* * *
**Minor Comments**

*[RC 2.03] L63 add Otto et al. (2022) to the list for completeness.*

**[AR 2.03]** This reference will be added.
* * *
*[RC 2.04] L147ff – Check the phrasing here with respect to the term "mean bedrock topography". Previously you generated the bedrock topography from the ice thickness models, now you go the other way…this does not make sense. I guess here you simply use the mean ice thickness model and not the bedrock topography. This would be in accordance to the Huss and Hock (2012) approach..*

**[AR 2.04]** The two formulations are equivalent but we agree that the wording might cause unnecessary confusion. We will reword into:

**Lines xx-xx:** "The current ice thickness distribution is taken from Grab et al. (2021), and GloGEM is applied to all glaciers of the SGI2016 with a glacier-specific calibration based on observed ice volume changes between 2000 and 2019 (Hugonnet et al., 2021)."
* * *
**[RC 2.05]** *L154 – replace or with for.*

**[AR 2.05]** Sure, this typo will be corrected.
* * *
**[RC 2.06]** *L188/189 – Erodibility is also affected by bedrock lithology. Sediment availability is equally important with respect to the tools required for abrasion. The former could probably not be accounted for here, while the latter is somehow represented by your consideration of headwall erosion. Please mentions these in the text.*

**[AR 2.06]** We perfectly agree and will mention bedrock lithology as an important, controlling mechanism. See also our reply to **RC 1.19**.
* * *
**[RC 2.07]** *L192 and L206ff – glacial and periglacial erosion….I would suggest to term this part solely periglacial or better headwall erosion (like you do in figure 5B), since you refer to the headwall area here only. Glacial erosion is represented by the approximation of abrasion in (1). Headwall erosion would include both processes, periglacial and feedbacks by glacial erosion.*

**[AR 2.07]** We agree and will only speak of "headwall erosion".
* * *
**[RC 2.08]** *L206 – consider adding some more recent references like: [Sanders et al. (2012) and/or Hartmeyer et al. (2020)]*

**[AR 2.08]** Both references are excellent suggestions and will be added.
* * *
**[RC 2.09]** *L386ff – In the methods section you described to quantify sediment infill rates in kg/m³ runoff. How do you relate these to erosion rates? (also relevant for figure 5B)*

**[AR 2.09]** Given the catchment-wide sediment infill rates (in $kg/m^3$, as described in the methods) and given the yearly runoff (in $m^3/a$, obtained from GloGEM), the total sediment yield (in kg/a) can be computed by simple multiplication. The specific erosion rates (in mm/a) are then obtained by assuming a given density for the eroded material (we assume $2800 kg/m^3$) and dividing by the total catchment area (which is known). We will add a short version of this explanation in the caption of Figure 5:

**Fig. 5, caption:** "[…] Numbers in the upper part of the panel show (i) sediment concentrations in the runoff […], and (ii) modelled catchment erosion rates. Both quantities are evaluated over 20-year time steps, and the latter quantity is obtained by assuming an average density for eroded materials of 2800 $kg\ m^{-3}$ and by dividing the catchment-wide sediment yield (now expressed as $m^3\ yr^{-1}$) by the catchment area."
* * *
**[RC 2.10]** *Figure 1 A: (1) rename the legend items...it seems like you depict the total deglaciated area and not the total lake area as described in the figure caption. (2) What does "1e6" represent at the upper left and lower right corners?*

**[AR 2.10]** (1) This was indeed a mistake and the legend will be corrected. (2) "1e6" was standing for "$10^6$", which was needed for the units of the coordinate system. We recognize that the information must have been confusing, and will amend that in the revised figure. The figure as it will appear in the revised manuscript is shown in our answer to **RC1.07**.

---

## Editor Decision (ED1)

[revised manuscript text omitted]